# Health Care for Refugees in Europe: A Scoping Review

**DOI:** 10.3390/ijerph19031278

**Published:** 2022-01-24

**Authors:** Anna Christina Nowak, Yudit Namer, Claudia Hornberg

**Affiliations:** 1School of Public Health, Bielefeld University, 33615 Bielefeld, Germany; 2Department of Epidemiology and International Public Health, School of Public Health, Bielefeld University, 33615 Bielefeld, Germany; yudit.namer@uni-bielefeld.de; 3Department of Sustainable Environmental Health Sciences, Faculty of Medicine, Bielefeld University, 33615 Bielefeld, Germany; claudia.hornberg@uni-bielefeld.de

**Keywords:** adult refugees, high-income European countries, access to health care, barriers and facilitators

## Abstract

Background: Accessing and using health care in European countries pose major challenges for asylum seekers and refugees due to legal, linguistic, administrative, and knowledge barriers. This scoping review will systematically describe the literature regarding health care for asylum seekers and refugees in high-income European countries, and the experiences that they have in accessing and using health care. Methods: Three databases in the field of public health were systematically searched, from which 1665 studies were selected for title and abstract screening, and 69 full texts were screened for eligibility by the main author. Of these studies, 44 were included in this systematic review. A narrative synthesis was undertaken. Results: Barriers in access to health care are highly prevalent in refugee populations, and can lead to underusage, misuse of health care, and higher costs. The qualitative results suggest that too little attention is paid to the living situations of refugees. This is especially true in access to care, and in the doctor-patient interaction. This can lead to a gap between needs and care. Conclusions: Although the problems refugees and asylum seekers face in accessing health care in high-income European countries have long been documented, little has changed over time. Living conditions are a key determinant for accessing health care.

## 1. Introduction

Asylum application numbers peaked in 2015 and 2016, when more than 2.5 million people applied for asylum in the European Union (EU) [1]. The high number of refugees and asylum seekers has challenged health care systems as health needs accompanied inequity in health care opportunities [2]. This renders migrant and refugee health a political public health topic [3].

It is difficult for refugees to access health care and for health care professionals to provide adequate care due to legal barriers, language barriers, administrative barriers, a lack of information and knowledge of the health care system, and breaks in the continuity of care [4]. When migration trajectories across Europe are considered, there are specific challenges in arrival, transit, and destination countries, such as a lack of coordination between health care providers in arrival countries, a short length of stay in transit countries, and missing social support in destination countries [5]. In particular, it is hard for refugees and asylum seekers to receive specialised care whilst in different migration phases [4,5].

As Lebano et al. (2020) [5] explained in their scoping review on migrants’ access to healthcare in nine European countries, research regarding this issue is scarce, and situations are difficult to compare. The scarcity of evidence and heterogeneity in structures make it even more important to examine the perspective of refugees and asylum seekers on access to health care. How refugees and asylum seekers experience access to care, and what facilitating factors and barriers they perceive, will be elicited in this scoping review. Therefore, the existing evidence will be presented to provide a narrative description of the complex and heterogeneous health care situations of refugees and asylum seekers in high-income European countries. For this purpose, the following questions were investigated:How do refugees and asylum seekers use health care in high-income European countries, and what barriers do they face?How do refugees experience access to health care in high-income European countries?

## 2. Materials and Methods

This review was conducted according to the PRISMA Extension for Scoping Reviews Checklist [6]. No review protocol was published beforehand. The literature search was conducted in the databases PubMed, CINAHL and PsycInfo in April 2017. The search was then updated in the PubMed database in August 2021. The following search terms were used: refugee, asylum seeker, health services accessibility, access to health care, delivery of health care, and adult (cf. Table 1). MeSH terms and/or simple search terms were combined for the search. The terms were linked with the Boolean operators AND and OR. An overview of the search terms is presented in additional file 1. In addition to a database search, studies were included according to the snowball principle (e.g., via bibliographies). No grey literature was included.

Screening was carried out by the first author only. After excluding duplicates, 1665 studies were included in the screening. After screening the titles, *n* = 158 were considered for further analysis. After screening the abstracts, *n* = 69 potentially relevant studies were identified, of which *n* = 44 were included in the present review after reading the full text (cf. Figure 1).

Studies with adult refugees and asylum seekers were included according to the provisions of the Geneva Refugee Convention or foreigners’ law equivalents. The groups of pregnant and minor refugees were excluded, as they are persons with special care needs. In order to ensure the greatest possible comparability, only studies that represent the situation in the European Region were included. Although the health care systems and residence regulations differ, there are common values between the participating states which have been established, for example, in the EU Charter of Fundamental Rights. These include the right to life in Article 2, the right of access to preventive health care in Article 35, or the right to non-discrimination in Article 21.

The analysis included both qualitative and quantitative studies in English and German published from 1992 onwards, due to the conclusion of the Maastricht Treaty and thus the increasing number of refugees and asylum seekers in European countries, and the associated challenges for health and social care throughout that time.

## 3. Results

Twenty-four quantitative studies, sixteen qualitative studies, and four studies with a mixed-method design were included. The most frequently reported barriers were language barriers (*n* = 19 studies) and the desire for interpreter care (*n* = 3), as well as information barriers (*n* = 14). Furthermore, administrative (*n* = 10) and financial barriers (*n* = 7) make access to the care system more difficult for refugees and asylum seekers. The selected studies are summarised according to the research questions in a narrative way. Table 2 presents an overview of the included studies.

### 3.1. Descriptive Results on Utilisation and Access Barriers

Twenty-two studies reported results on health care access for refugees and asylum seekers, such as formal and bureaucratic barriers in reception countries [7,8,9,10,11,12,13], during flight [4,14], and the influence of length of stay and legal status on the use of health care services [7,8,15,16,17,18]. Five studies [18,19,20,21,22] took into account the use of health care by groups under vulnerable conditions, such as refugees with mental illnesses. Two studies evaluated the financial burden of the health care systems used by refugees and asylum seekers [23,24]. Three other studies compared two access models for refugees in Germany [25,26,27].

**Table 2 ijerph-19-01278-t002:** Overview of the included studies [4,7,8,9,10,11,12,13,14,15,16,17,18,19,20,21,23,24,25,26,27,28,29,30,31,32,33,34,35,36,37,38,39,40,41,42,43,44,45,46,47,48,49,50].

Authors	Publication Year	Country	Setting	Study Design
Bauhoff and Göpffarth [24]	2018	Germany	Cost analysis	Quantitative
Bhatia and Wallace [8]	2007	UK	Primary Care	Qualitative
Bhui et al. [15]	2006	UK	Inpatient psychiatric care	Quantitative
Bianco et al. [31]	2015	Italy	Health care by NGOs	Quantitative
Biddle et al. [32]	2019	Germany	Inpatient and outpatient care	Quantitative
Bischoff et al. [40]	2003	Switzerland	Inpatient care	Quantitative
Blöchliger et al. [50]	1998	Switzerland	Primary Care	Quantitative
Boettcher et al. [18]	2021	Germany	Psychiatric care	Quantitative
Borgschulte et al. [29]	2018	Germany	Primary care	Mixed-Method
Bozorgmehr et al. [30]	2015	Germany	Primary Care	Quantitative
Chiarenza et al. [4]	2019	European countries	Transit	Mixed-Method
Cignacco et al. [28]	2018	Switzerland	Sexual and reproductive care	Mixed-Method
Claassen & Jäger [26]	2018	Germany	Primary Care	Quantitative
Fang et al. [12]	2015	UK	Primary Care	Qualitative
Feldmann et al. [47]	2007a	Netherlands	Primary Care	Qualitative
Feldmann et al. [49]	2007b	Netherlands	Primary Care	Qualitative
Führer et al. [19]	2020	Germany	Psychiatric care	Quantitative
Gerritsen et al. [34]	2006	Netherlands	Primary Care	Quantitative
Hahn et al. [10]	2020	Germany	Inpatient and outpatient care	Qualitative
Jäger et al. [27]	2019	Germany	Primary care	Quantitative
Jensen et al. [46]	2014	Denmark	Inpatient psychiatric Care	Qualitative
Kang et al. [13]	2019	UK	Primary Care	Qualitative
Klingberg et al. [35]	2020	Switzerland	Emergency care	Quantitative
Kohlenberger et al. [33]	2019	Austria	Inpatient and outpatient care	Quantitative
Laban et al. [16]	2007	Netherlands	Primary care	Quantitative
Lamkaddem et al. [38]	2014	Netherlands	Psychiatric Care	Quantitative
Maier et al. [23]	2010	Switzerland	Cost analysis	Quantitative
Mangrio et al. [43]	2018	Sweden	Inpatient and outpatient care	Mixed-Method
Mårtensson et al. [44]	2020	Sweden	Health care information	Qualitative
Melamed et al. [20]	2019	Switzerland	Psychiatric care	Qualitative
Norredam et al. [7]	2005	EU countries	Comparative study	Quantitative
O’Donnell [39]	2007	UK	Primary Care	Qualitative
O’Donnell [48]	2008	UK	Primary Care	Qualitative
Razavi et al. [45]	2011	Sweden	Long term health care	Qualitative
Riza et al. [37]	2020	EU	Inpatient and outpatient care	Quantitative
Schein et al. [42]	2019	Norway	Inpatient and outpatient care	Qualitative
Schneider et al. [11]	2015	Germany	Inpatient and outpatient care	Quantitative
Spura et al. [9]	2017	Germany	Inpatient and outpatient care	Qualitative
Toar et al. [21]	2009	Ireland	Psychiatric care	Quantitative
Van Loenen et al. [14]	2017	EU	Primary Care	Qualitative
Wångdahl et al. [36]	2018	Sweden	Health literacy	Quantitative
Wenner et al. [25]	2020	Germany	Inpatient and outpatient care	Quantitative
Wetzke et al. [17]	2018	Germany	Primary Care	Quantitative
Zander et al. [41]	2013	Sweden	Inpatient and outpatient care	Qualitative

Refugees and asylum seekers experience different access barriers to health care during flight through Europe [4,14]. In the countries of arrival and transit, the high number of asylum seekers, short-term stays, and uncoordinated health care can negatively impact access to, and quality of, care. A lack of patient records and information on the health status of patients during flight and in reception countries [14,28,29], as well as the involvement of different actors and organisations [4,14], lead to a lack of equity of need, duplication of treatment, and overlaps in care. In the destination countries, asylum seekers face a multitude of legal, financial, and administrative barriers to access [4]. Access to the health care system has primarily been subject to formal legal barriers [7]. In 10 out of 23 surveyed countries in the EU in 2004, there were access barriers to the health care system. Such countries include Austria, Denmark, Estonia, Finland, Germany, Hungary, Luxembourg, Malta, Spain and Sweden. With the exception of Austria, asylum seekers were only entitled to emergency health care. Although health care access has changed in some countries after the refugee migration in 2015/2016, this has not been systematically evaluated yet.

More recent studies have evaluated the utilisation rates of refugees and asylum seekers in host countries. These range from 1% to 90%, depending on physician specialty. In literature, utilisation rates of 52–85% for GPs [30,31,32], 22.4–42.5% for specialist care [30,32,33], and from 1% to 15.5% for psychotherapeutic care were seen [18,19,30]. Differences can be partly explained by gender [30,33,34], length of stay [16,17], country of origin [24,33], age [34] and low health literacy [35,36]. Differences are probably also due to the living and housing situation, as well as legal status, and formal restrictions in access to care. In comparison to reference populations, a lower use of GPs but higher use of hospitals and emergency care were seen [11,24], which can lead to higher expenditures [23,24]. Unmet health care needs are highly prevalent, and range from 20% to 45.2% [18,30,32,37]. Refugees and asylum seekers with mental illness found it especially difficult to receive adequate care [18,19,21,23,38]. In addition, a lack of culturally and gender-sensitive systems inhibits access to health care [10,28].

By reducing bureaucratic barriers, the use of health care can be improved, as has been shown, for example, through the introduction of new care models in Germany [25,26,27]. In addition, needs-based accommodation and social support can play a major key in access to health care [22,31].

### 3.2. Lived Experiences of Asylum Seekers and Refugees Using Health Care in Host Countries

Qualitative findings can be used to explain the experiences refugees and asylum seekers have and the barriers that they face in transit through Europe, and in host countries, in more detail. Language challenges are identified as key barriers to accessing and using the care system [8,10,12,13,39,40], due to experiences of dependency to understand, and to be understood [41]. Difficulties in obtaining a qualified interpreter are apparent. Interruptions in care can occur when interpreters are not available. At times, the relationship with the interpreter is characterised by mistrust [8,12,20,39]. This can lead to misunderstandings, and incorrect diagnoses in treatment situations. From the perspective of refugees and asylum seekers, it is important that interpreters know medical terminology and are culturally competent [12]. In addition, in two studies, a strong desire for continuous support from the same interpreter was found [8,39]. One study highlights the importance of language acquisition to the experience of autonomy and control in the treatment situation [42].

“Waiting” plays a key role—both in asylum seekers’ lives, and in health care [12,20,42,43]. For example, refugees describe having to wait a long time for doctor’s appointments. This is particularly important when symptoms are interpreted differently, as O’Donnell et al. (2007) [39] describe. For example, flu symptoms may, in some countries of origin, indicate a life-threatening illness, whereas in host countries they indicate little need for medical treatment. The discrepancy between subjective and medical needs, and the lack of familiarity with the spectrum of illnesses in the destination countries [44], can potentially lead to late access to care for serious health problems [12], or to emergency room visits for non-critical illnesses [39]. From the refugees’ perspective, concrete and explanatory health information made available from a variety of sources could help to strengthen knowledge on the health care system [44].

The complexity of health care processes and procedures can present challenges [45,46]. This is particularly evident in transit through Europe [14], when unfamiliarity with illness and treatment strategies can result in a discrepancy between expectations of care, and actual care [45]. This can lead to care disruptions [41]. As a result, refugees may feel uncertain as to how they should behave in the system, and what is expected of them as patients [45].

Six of the studies highlight the importance of friends, family, and social organisations in negotiating health care [8,20,38,42,46,47]. Social support is seen as the most important component for experiencing health and illness. The breakdown of trusting contacts in destination countries can affect mental health and lead to the non-utilisation of care [20]. Where support is available to access the health care system, e.g., an asylum support nurse in the accommodation, feelings of being welcomed are reported [39]. At the same time, the accommodation situation can be experienced as a barrier to accessing health care, especially when having to address the manager to ask for medical treatment [20,42]. A similar situation is reported by Spura et al. (2017) [9] and Hahn et al. (2020) [10] for the issuing of treatment vouchers at the social welfare office for asylum seekers in Germany.

The legal situation can also be an influencing factor. For example, Fang et al. (2015) [12] report that health care is not used because participants are afraid of consequences to their legal status. Schein et al. (2019) [42] note that the experience with the Norwegian health care system depends on the length of stay and the residence status. Uncertainty in the ongoing asylum procedure overlays the entirety of an asylum seekers’ life, leaving little space for healthcare. In addition, the interviewees feel “invisible” in the care system [42], as they do not possess a national insurance number, or “left out” by being an asylum seeker [39]. A rejected asylum decision can lead to a sudden lack of entitlements to health care, and thus the experience of loss of resources [12].

Societal conditions lead to experiences of hostility in health care through which refugees experience themselves as a “burden” for the system [8]. They may then withdraw from the health care system out of fear of stigmatisation. Moreover, refugees are often treated as representatives of refugees as a group, and not seen as individual patients [47]. Labelling as an “asylum seeker” or “refugee” leads to experiences of othering [12], and to possibly false assumptions about health status [47]. This creates feelings of difference and exclusion [39,48].

Too little attention is paid to the complex health needs and realities of refugees’ lives [12,45,46]. This can lead to perceived excessive medicalisation [45,46] because life experiences are not sufficiently acknowledged in the health care system. In addition, re-traumatising experiences can be triggered, as described by an interviewee in the study by Jensen et al. (2014) [46], who experienced flashbacks relating to her prison stay in her home country by being placed in a closed psychiatric ward.

Jensen et al. (2014) [46] provide insight into the specific care needs of refugees and asylum seekers. In qualitative interviews with native Danes, migrants, and refugees about access to and the course of treatment in psychiatric care in Denmark challenges were seen for all participants. These challenges in accessing the health system and in the course of treatment are characterised by rejection, interruptions, and transitions. At the same time, unique challenges for refugees and asylum seekers arise as there is not always a social network present to help access health care easily or to ensure childcare in the case of an inpatient stay.

The promoting factors mentioned include a trusting relationship with the attending doctor [8,48], adequate provision of interpreters [9,12], respectful interaction [12,49], comprehensive information about the health care system [12,39,44], an individual approach to health problems [47,49], and a good social network [41]. Adequate intercultural training of doctors could also reduce existing health care problems [50].

## 4. Discussion

This review reveals that health care for refugees in high-income European countries is restricted by a multitude of access barriers. Institutional and legal barriers to access especially impede the integration of refugees into regular care. Language and information barriers can lead to mistreatment, inadequate care provision, and underusage of (ambulatory) services, especially for patient groups in vulnerable conditions. All reported barriers lead to challenges for health care professionals and patients, as well as unmet care needs.

In this scoping review, we considered papers from 1992 onwards. There have been few changes in access to health care for refugees and asylum seekers over time. In particular, when comparing the data before and after the 2015 refugee movement, we can identify the same care problems, despite the fact that the following care problems have been well known for a long time: restrictions in care, language and information barriers, lack of funding for interpreters, no adequate care pathways, and lack of culturally sensitive care services. Overall, needs-based care for refugees in high-income European countries is rare, and health care does not correspond adequately. Recommendations for need-based care for refugees can be found in the literature, and can be drawn from our scoping review [4,5,51,52]: removal of legal restrictions, funding and provision of culturally sensitive and trained interpreters [53,54], strengthening health literacy [55], early screening for mental illness [52], low-threshold access to care [37], and intercultural training to reduce stereotyping and avoid discrimination [49].

The included qualitative studies provide new insights into the experience of health care in host countries. It can be seen that access to health care as a determinant of health [56] is strongly influenced by life circumstances. Postmigration stressors, such as legal status [42], hostile conditions in host countries [8], or lack of social network [20], play a key role here. This is especially important for refugees because health can be an influencing facilitator of integration, as well as the outcome of successful integration [57,58]. However, qualitative data reveal that refugees often find themselves in ongoing dependency relationships, for example, due to restrictions in access to the health care system by authorities, or facility managers, having to rely on adequate translation by the interpreter, or in relation to their residence status. This leads to feelings of disempowerment and exclusion in access to and use of health care [39]. Health care professionals should therefore be aware of the individual situations of refugees and asylum seekers. However, it has been shown that life experiences are not sufficiently accounted for in the treatment situation, which can be associated with negative health consequences [46]. In some cases, discriminatory and exclusionary practice due to the homogenisation of refugees and asylum seekers is evident [47]. Therefore, culturally sensitive and individualised care is necessary to empower refugees and asylum seekers to make adequate health decisions for themselves. For this, it is even more important to take the living conditions into account. For example, support for language acquisition early in their stay can strengthen the position of asylum seekers and refugees in health care. As shown by Schein et al. (2019) [42], improvements in language skills lead to experiences of autonomy and control in the treatment situation. Needs-based accommodation can also be beneficial [22]. Therefore, the access to and use of health care have to be seen in a broader socio-economic-legal context, with special emphasis on individuals’ life realities.

The overall aim of this scoping review was to map the literature and provide a broad overview of access to health care for refugees and asylum seekers in high-income European countries, including the perspectives of refugees and asylum seekers. Although the literature has been comprehensively presented, there have been methodological and content-related limitations. The literature search and screening were conducted only by the first author. Therefore, it may be that not all relevant studies have been included. Grey literature, which may have led to further insights in the research field, was excluded. Due to the inclusion of different study types with heterogenous outcome parameters, no quality assessment was carried out. Therefore, a potential bias of studies must be taken into account.

Even though we only included refugees and asylum seekers, they are a heterogenous study population with diverse cultural and sociodemographic backgrounds. The impact of gender or socioeconomic background on health care needs and associated use and access to health care services was not considered. As shown in this review, reception conditions differ in high-income European countries, and so it was sometimes difficult to generalize results. In general, however, barriers to the access to health care for refugees and asylum seekers are evident across national borders, and they are vocalized by refugees and asylum seekers in detail.

## 5. Conclusions

Access to health care for refugees and asylum seekers in high-income European countries is challenging and has scarcely changed over time. In addition to removing legal, linguistic, formal, and organisation barriers, the living conditions of refugees should be given greater consideration in the provision of care. On the one hand, they influence the development of health and illness directly. On the other hand, an improvement in living conditions and liberation from dependencies can lead to a sense of agency and experiences of empowerment in health care.

## Figures and Tables

**Figure 1 ijerph-19-01278-f001:**
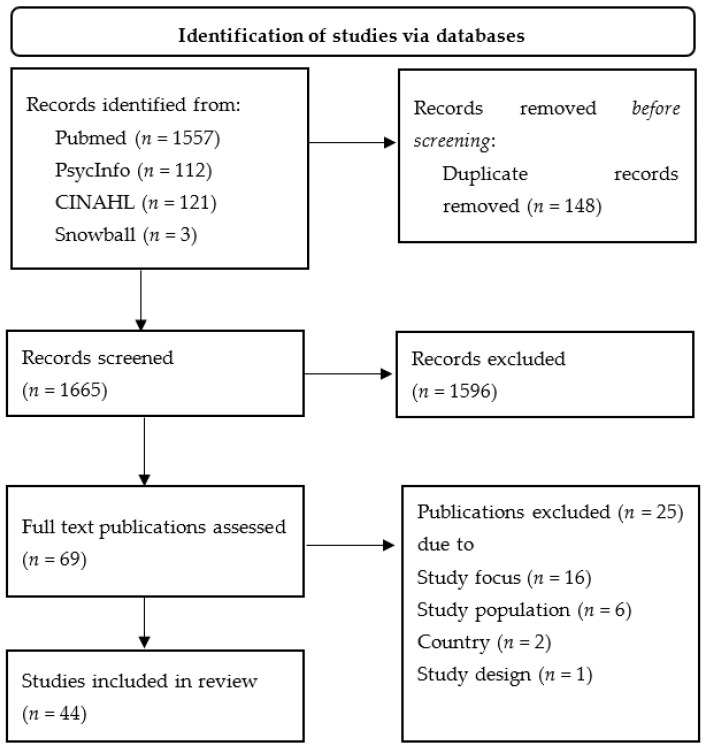
PRSIMA flow chart.

**Table 1 ijerph-19-01278-t001:** Inclusion and exclusion criteria for the literature search.

Inclusion Criteria	Exclusion Criteria
Refugees and asylum seekers over 18 years of age Region: high-income European countries Qualitative and quantitative studies Year: from 1992	Pregnant refugees Internally displaced people Individual case analyses, diagnostic and therapeutic studies, grey literature, systematic reviews

## Data Availability

All data generated or analysed during this study are included in the published article.

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
