# Peer review of "Health Care for Refugees in Europe: A Scoping Review"

_ijerph, 2022, doi:10.3390/ijerph19031278_

Round 1

Reviewer 1 Report

This paper reports on an important public health issue in Europe. A literature review on health care of refugees has the potential to generate new insights based on several recent studies. Yet, I think that this promise has not come true, due to various reasons.

  1. The objective is not clearly stated. The abstract states that it aims, very broadly, “to describe literature on health care for asylum 14 seekers and refugees in high-income European countries”. The first research question is hard to understand: “How does access on macro-, meso- and micro-level work for refugees and asylum 59 seekers …”.
  2. According to the introduction, the novel feature of this paper may be in “to examine the perspective of refugees and asylum seekers on access to health care.” Yet, this insiders perspective is not clear from the results. Many results are about quantitative relationships as observed with statistics.
  3. The very start of the introduction lists the several types of barriers that are already known to play a role (page 1, lines 37-40). The results section of this paper adds very little to this list. The same barriers are mentioned, but each time only in a few words. One does not get a picture regarding WHICH specific factors and processed play a role, and HOW they affect refugees’ experiences with health care.
  4. As the authors recognize themselves, two main limitations are that the screening is done by only one author, and that there is no assessment of the quality of the studies. The latter is a main problem given that the vast majority are published in low ranking journals.
  5. The ordering of the results section is not clear. It is unclear, for example, why the description of levels of health care utilization is done under “meso” factors. One would expect such a description of levels of utilization before the underlying factors at macro, meso and micro levels are discussed.
  6. Studies on levels of utilization appear to give widely different results. There is no attempt of the authors to identify some general patterns among all the varied results. Instead, it is left up to the reader to try to generalize from the (possibly arbitrary) listing of findings. I find this confusing.
  7. The selection of results seem arbitrary and does not logically derive from e.g. the second research question. It is unclear, for example, why so much attention is give to one specific German intervention (page 6, line 168-181) or the difference between LGBTIQ refugees versus other refugees (page 6, line 204-212).
  8. The Conclusion do not formulate new insights that derive from this systematic review, but instead repeats what has been written before.
  9. Reference 54 and 55 are grey literature. These are studies co-authored by the first author. I think that they should be excluded, given that grey literature was explicitly excluded according to the search strategy.

Author Response

Dear reviewer, thank you very much for the helpful advice on our manuscript. We have taken your comments into account and submitted the revised manuscript. Please find our point-to-point response in the attached file.

Reviewer 2 Report

Congratulations to the authors for a comprehensive literature review and analysis of the issues associated with health care for refugees in Europe. A few typos need to be fixed:

Sentence 209 please change 'higer' to 'higher'.

Sentence 210 please correct 'accommodation'  and 'queer-'.

Sentence 257 please separate 'particularlyfor'

Sentence 287 spacing at the beginning of the second sentence is required.

Sentence 348 please delete one of the 'changes'

Sentence 374 please split 'implementinghealth'

Author Response

(The authors gave the same response as above.)

Reviewer 3 Report

This paper presents a good overview of health care barriers for refugees and asylum-seekers in Europe. However, there is little new information derived from this review - these barriers were already widely known.

Perhaps the most valuable finding of this study is that little has changed in health care access for these populations in thirty years. However, the authors then make the same recommendations for change that have been being made for the same thirty years. The manuscript should consider much more deeply the socio-political-economic-structural change strategies needed that differ from the "same old, same old." 

Author Response

(The authors gave the same response as above.)

Round 2

Reviewer 3 Report

The authors have satisfactorily addressed the previous comments.